# Flexible Artificial Ag NPs:a–SiC_0.11_:H Synapse on Al Foil with High Uniformity and On/Off Ratio for Neuromorphic Computing

**DOI:** 10.3390/nano14181474

**Published:** 2024-09-10

**Authors:** Zongyan Zuo, Chengfeng Zhou, Zhongyuan Ma, Yufeng Huang, Liangliang Chen, Wei Li, Jun Xu, Kunji Chen

**Affiliations:** 1School of Electronic Science and Engineering, Nanjing University, Nanjing 210093, China; dg1923064@smail.nju.edu.cn (Z.Z.); 602022230060@smail.nju.edu.cn (C.Z.);; 2Collaborative Innovation Center of Advanced Microstructures, Nanjing University, Nanjing 210093, China; 3Jiangsu Provincial Key Laboratory of Photonic and Electronic Materials Sciences and Technology, Nanjing University, Nanjing 210093, China

**Keywords:** memristor, flexibility, multilevel resistive switching, uniformity

## Abstract

A neuromorphic computing network based on SiC_x_ memristor paves the way for a next-generation brain-like chip in the AI era. Up to date, the SiC_x_–based memristor devices are faced with the challenge of obtaining flexibility and uniformity, which can push forward the application of memristors in flexible electronics. For the first time, we report that a flexible artificial synaptic device based on a Ag NPs:a–SiC_0.11_:H memristor can be constructed by utilizing aluminum foil as the substrate. The device exhibits stable bipolar resistive switching characteristic even after bending 1000 times, displaying excellent flexibility and uniformity. Furthermore, an on/off ratio of approximately 10^7^ can be obtained. It is found that the incorporation of silver nanoparticles significantly enhances the device’s set and reset voltage uniformity by 76.2% and 69.7%, respectively, which is attributed to the contribution of the Ag nanoparticles. The local electric field of Ag nanoparticles can direct the formation and rupture of conductive filaments. The fitting results of I–V curves show that the carrier transport mechanism agrees with Poole–Frenkel (P–F) model in the high-resistance state, while the carrier transport follows Ohm’s law in the low-resistance state. Based on the multilevel storage characteristics of the Al/Ag NPs:a–SiC_0.11_:H/Al foil resistive switching device, we successfully observed the biological synaptic characteristics, including the long–term potentiation (LTP), long–term depression (LTD), and spike–timing–dependent plasticity (STDP). The flexible artificial Ag NPs:a–SiC_0.11_:H/Al foil synapse possesses excellent conductance modulation capabilities and visual learning function, demonstrating the promise of application in flexible electronics technology for high-efficiency neuromorphic computing in the AI period.

## 1. Introduction

With the advent of the artificial intelligence era, the explosive growth of data and the increasing demand for efficient human–computer interaction are driving research in neuromorphic networks that mimic the information processing of the human brain. Achieving efficient and parallel artificial neural networks is considered an effective way to overcome the traditional von Neumann bottleneck [1,2,3]. Neuromorphic networks can simulate the brain’s efficient distributed storage and computation, approximating the behavior of biological neural networks, and have become a hot topic in international research [4,5,6]. Mimicking the structure and function of biological synapses, constructing artificial synaptic devices with biological synapse functions is a key step in developing efficient artificial neural networks and serves as a fundamental component for intelligent human–computer interaction interface [7,8,9]. In particular, the development of artificial intelligence and the Internet of Things has fueled market demand for lightweight and portable consumer electronics. Flexible devices with good bending stability are gaining widespread attention due to their potential applications in wearable electronics, biomimetic sensors, and brain-inspired chips [10,11]. In fact, the mechanical flexibility of neuromorphic devices used to simulate biological neurons and synapses is crucial for accurately mimicking synaptic functions, as neurons and synapses in biological systems are typically soft and elastic, adapting to various forms of mechanical deformation [12,13]. Thus, achieving flexibility in artificial synapses is an inevitable trend in future development, which also means that the resistive random-access memory (RRAM) used to build artificial synapses needs to be flexible.

In recent years, many researchers have focused on the design and study of flexible RRAM [14,15,16,17,18]. With the deepening research into flexible RRAM and the surge in demand for constructing artificial neuromorphic networks, there has been a new wave of research into flexible artificial synapses based on RRAM [19,20]. Flexible artificial synapse devices, with their advantages of being lightweight, inexpensive, and bendable, are driving the development of semiconductor memory and flexible electronics technology. However, the choice of flexible substrates remains a challenge in the development of flexible devices. Existing flexible substrates primarily consist of organic polymers like polyethylene terephthalate (PET) [21,22,23], polyethersulfone (PES) [24,25,26], and polyimide (PI) [27,28,29]. These materials have good mechanical flexibility, can withstand repeated bending and twisting, and possess stable physical and chemical properties. However, their low glass transition temperature, typically below 200 °C, limits their further application. Additionally, when growing thin films on organic polymer flexible substrates, issues such as weak adhesion and easy delamination often arise. Especially after repeated bending, leading to a significant increase in film resistivity, which directly impacts the performance of RRAM, reduces its cycle life, and causes premature device failure. Although PI substrates can tolerate high temperatures up to 400 °C, their high cost makes them difficult to use in device commercialization. Therefore, finding a substrate that is both high-temperature resistant and flexible is crucial for developing new types of flexible artificial synapse devices.

Here, we report that a flexible artificial synaptic device based on Ag NPs:a–SiC_0.11_:H memristor can be constructed by utilizing aluminum foil as the substrate. Aluminum foil, as an ultra-flexible material, is thinner, more flexible, and easier to bend compared to polymer substrates. Its temperature resistance reaches up to 660 °C, which is not affected by the high-temperature environment during film growth. Additionally, aluminum foil has high electrical conductivity, allowing it to serve as both a substrate and an electrode. This not only simplifies the device fabrication process but also reduces the device volume, which is advantageous for high-density storage devices. The device exhibits stable bipolar resistive switching characteristic even after bending 1000 times, displaying excellent flexibility and uniformity. Furthermore, an on/off ratio of approximately 10^7^ can be obtained. It is found that the incorporation of silver nanoparticles significantly enhances the device’s set and reset voltage uniformity by 76.2% and 69.7%, respectively, which is attributed to the contribution of the Ag nanoparticles. The local electric field of Ag nanoparticles can direct the formation and rupture of conductive filaments. Based on the multilevel storage characteristics of the Al/Ag NPs:a–SiC_0.11_:H/Al foil resistive switching device, we successfully observed biological synaptic characteristics, including long–term potentiation (LTP), long–term depression (LTD), and spike–timing–dependent plasticity (STDP). The flexible artificial Ag NPs:a–SiC_0.11_:H/Al foil synapse possesses excellent conductance modulation capabilities and visual learning function, demonstrating the promise of application in flexible electronics technology for high-efficiency neuromorphic computing in the AI period.

## 2. Materials and Methods

Using tweezers, the clean aluminum foil was attached to the silicon substrate and flattened as much as possible. Afterward, the silicon substrate wrapped with aluminum foil was placed into the PECVD chamber for the growth of a–SiC_0.11_:H film with SiH_4_ to CH_4_ as the precursor, using a gas flow ratio of 5:1. The growth temperature was 250 °C. The thickness of a–SiC_0.11_:H film was 30 nm. After taking the samples out of the chamber, the 10 nm diameter Ag nanoparticles (Ag NPs) were uniformly spin-coated onto the film using the spin-coating method. A circular hole mask was used to obtain a circular electrode, and the aluminum top electrodes were grown using thermal evaporation technology. Finally, a flexible Al/Ag NPs:a–SiC_0.11_:H/Al foil memory device was obtained, as displayed in Figure 1a. Compared to the transfer or peeling techniques used for preparing polymer flexible substrates [30,31], this preparation method is not only cost-effective, simple, and convenient but also has a high success rate. Cross-section high-resolution transmission electron microscopy (X-HRTEM, JEOL 2100F electron microscope, JEOL, Kyoto, Japan) was used to analyze the microstructure of the Al/Ag NPs:a–SiC_0.11_:H/Al foil device. A scanning electron microscope (SEM, Fei NovaNanoSEM 650, FEI, Hillsboro, OR, USA) was used to characterize the morphologies of the samples. The chemical compositions of the a–SiC_0.11_:H films were obtained through X-ray photoelectron spectroscopy (XPS) using the PHI 5000 Versa Probe (ULVAC, Kanagawa, Japan). The electron spin resonance (ESR) spectrum was measured in a Bruker EMX–10/12 (Bruker, Billerica, MA, USA) system within a temperature range of 2–290 K, and Fourier transform infrared (FTIR) spectrum testing was performed in the NEXUS870 (ThermoFisher, Waltham, MA, USA) system. The Agilent B1500A semiconductor analyzer (Agilent Inc., Santa Clara, CA, USA) was used to test the electrical characteristics of the Al/Ag NPs:a–SiC_0.11_:H/Al foil device. The Lake Shore CRX–4K system (Lake Shore Inc., Westerville, OH, USA) was used for measuring the electronic characteristics in a variable temperature environment.

## 3. Results and Discussion

The X-HRTEM image of the Al/Ag NPs:a–SiC_0.11_:H/Al foil device is shown in Figure 1d. A laminated structure with Ag NPs:a–SiC_0.11_:H as the middle layer is clearly observed. The SiC_0.11_:H layer exhibits an amorphous state. Figure 1b shows the SEM image of Ag NPs:a–SiC_0.11_:H on the surface of Al foil. The flexible Al/Ag NPs:a–SiC_0.11_:H/Al foil RRAM device exhibits a certain degree of flexibility, as shown in the inset of Figure 1c. To test the electrical performance of the device in a flat state, the Al/Ag NPs:a–SiC_0.11_:H/Al foil RRAM device was attached to a silicon substrate, as illustrated in Figure 1c. Before electric measurement, we used XPS to analyze the composition of the a–SiC_0.11_:H films, as shown in Figure 2a. The XPS narrow scan spectra were deconvoluted by fitting the data with a number of Gaussian peaks. Two main peaks corresponding to Si–C (100.8eV) and Si–Si(99.2eV) were detected. The Si–Si bond indicates that excess uncreated Si exists in the a–SiC_0.11_:H films. In order to detect the trap centers distributed in the a–SiC_0.11_:H films, the temperature-dependent ESR spectra of the a–SiC_0.11_:H films was measured and is displayed in Figure 2b. An obvious resonance peak can be observed at room temperature, whose intensity gradually increases with the decrease in temperature. The value of g factor was 2.0027, which indicates the existence of Si dangling bonds (Si DBs) in the a–SiC_0.11_:H films. As presented in Figure 2c, the FTIR spectra of the as-deposited a–SiC_0.11_:H films shows that the absorption bands at 780 cm^−1^, 1000 cm^−1^, 2000 cm^−1^, and 2900 cm^−1^ corresponded to the Si–C stretching vibrations, Si–H wagging vibrations, Si–H stretching vibrations, and C–H stretching modes, respectively. Note that the intensity of either Si–H wagging or Si–H stretching is much stronger than that of Si–C and C–H bonds, which provides strong evidence that there are a number of Si–H bonds at the initial state.

During electrical testing, a positive voltage was applied to the top Al electrode, and a current limit of 10 mA was set to prevent the device from breaking down due to excessive current in the low-resistance state. Figure 2d displays the current–voltage (I–V) curves of the flat Al/Ag NPs:a–SiC_0.11_:H/Al foil device. The bipolar resistance switching behavior can be observed after continuous scanning with DC voltage. The red line in Figure 2d represents the voltage value of the device during the first set process, indicating the forming-free characteristics of the flat Al/Ag NPs:a–SiC_0.11_:H/Al device [32,33]. Most RRAM devices require a large forming voltage to create the necessary conductive channels in the resistance switching layer before achieving stable reversible switching between high- and low-resistance states. This prerequisite complicates the peripheral circuits of the RRAM device and results in increased power consumption, posing limitations on practical applications [34]. Our Al/Ag NPs:a–SiC_0.11_:H/Al foil devices with forming-free characteristics are crucial for achieving low-power, non-volatile memory applications. By applying a sequence of voltage cycles: 0V to 5 V, 5 V to 0 V, 0 V to −2 V, and −2 V to 0 V, the device transitions between high- and low-resistance states. The flat Al/Ag NPs:a−SiC_0.11_:H/Al device can switch from a high-resistance state (HRS) to a low-resistance state (LRS) under the voltage of 3 V. Multiple scan results show that the set voltage ranges between 2 V and 3 V, indicating a “sudden” set process. The reset curve displays several “steps,” where the output current significantly decreases under the negative voltages of −0.5 V, −0.8 V, and −1 V, indicating that the device is gradually reset to different HRS states. When the voltage changes from −1.4 V to −1.6 V, the device is almost completely reset, returning to the final HRS state. It was demonstrated that adjusting the reset voltage can achieve continuously tunable conductance. Thus, Al/Ag NPs:a−SiC_0.11_:H/Al foil devices have the capability of multi-level storage and the potential for simulating biological synaptic plasticity. Figure 2e shows the endurance characteristic of the flat Al/Ag NPs:a−SiC_0.11_:H/Al device after 1000 cycles. It is clear that the high- and low-resistance values fluctuate slightly during the 100 cycles of write and erase processes, maintaining an ultra-high on/off ratio of around 10^7^, demonstrating good stability and reliability. Figure 2f shows the retention performance of the device in high-/low-resistance states over a period of more than 10^4^ s. It was found that the storage window did not show significant changes compared to the initial test, demonstrating good retention performance.

For comparison with the Al/Ag NPs:a−SiC_0.11_:H/Al foil devices, we prepared Al/a−SiC_0.11_:H/P^+^−Si structured RRAM devices. Figure 3a,b shows the distribution of the set/reset voltages for these two types of devices after 50 cycles of operation. It is evident that the distribution range of set/reset voltages for the Al/Ag NPs:a−SiC_0.11_:H/Al foil device is more concentrated compared to that of the Al/a−SiC_0.11_:H/P^+^−Si device, indicating smaller switching voltage fluctuations for the Al/Ag NPs:a−SiC_0.11_:H/Al foil device, as displayed in Figure 3c. To further insight the variation of the set voltage and reset voltage after 30 cycles, the coefficient of variation was analyzed, which can be expressed by the ratio of the standard deviation (σ) and the mean value (u) of V_SET_ or V_RESET_ in absolute value. The variation coefficient of the high-resistance state and the low-resistance state for Al/Ag NPs:a−SiC_0.11_:H/Al foil devices were reduced 76.2% and 69.7%, respectively, in contrast with that of the Al/a−SiC_0.11_:H/P^+^−Si device. As reported in our previous work [35,36], the a-SiC_0.11_:H film contains a certain number of randomly distributed Si DBs, which is revealed in Figure 2b. They can combine with the new generation of Si DBs under the forward electric field to form the conductive filaments connecting the top and bottom electrodes, ensuring the existence of LRS. As evidenced in Figure 2c, the number of Si−H bonds is larger than that of Si−C bonds and C−H bonds. Production of Si DBs is induced by the broken Si−H bond due to the intensity of the electric field. Because the bonding energy of Si−H bond is lowest compared with other bond structures, such as Si−C bonds and C−H bonds, which are easier to break. Under the reverse voltage, H^+^ comes back to saturate the silicon dangling bonds, thereby breaking the conduction pathway and switching the device to the HRS state. Introducing silver nanoparticles into the device can enhance the local electric field due to the high curvature of the surface of Ag NPs, reducing the randomness of the formation and rupture positions of the conductive filaments based on silicon dangling bond and significantly improving device performance uniformity, as illustrated in Figure 3d,e. This is the reason why the Al/Ag NPs:a−SiC_0.11_:H/Al foil device has smaller switching voltage fluctuations. The double-logarithmic fitting of the set process for the flat Al/Ag NPs:a−SiC_0.11_:H/Al device is shown in Figure 3f. It is observed that the set process in the low-resistance state is a straight line with a fitted slope of 0.99 (approximately 1), exhibiting an I−V linear relationship. The I−V curve follows Ohm’s law in the low-resistance state. Because most of the silicon dangling bonds are filled, and the free carriers dominate the current transmission. To analyze the current transport mechanism in the high-resistance state, the I−V curve is shown in Figure 3g. According to the literature, the I−V relationship is dominated by the Poole–Frenkel (P−F) conduction mechanism [37,38]:(1)$$J\in Eexp(\frac{q\sqrt{qE/\pi \varepsilon_{0}\varepsilon_{d}}-q\Phi_{PF}}{kT})$$
where J is the current intensity, E is the electric field intensity, k is the Boltzmann constant, T is the temperature, ε_0_ is the vacuum permittivity, ε_d_ is the dielectric constant of the dielectric layer, and Φ_PF_ is the Schottky barrier. According to Formula (1), ln(I/V) has a linear relationship with Sqrt(V). As shown in Figure 3g, the fitting curve in the high-resistance state is a horizontal line in the voltage region below 0.4 V. The value of ln(I/V) has no dependance with sqrt(V), obeying Ohm’s law. In the voltage region above 0.4 V, the fitting curve is a linear line, indicating that the current transmission follows the P−F conduction mechanism. The reduced silicon DBs are responsible for the transmission, which is induced by the passivation of H^+^ ion. The electrons captured by Si DBs transmit directly from defect states to the conduction band following the P−F emission mechanism.

The Al/Ag NPs:a−SiC_0.11_:H/Al foil devices were adhered to glass cylinders with radii of 10 mm and 6 mm to examine the electric characteristics under different bending conditions, as shown Figure 4a. The I−V curves of the Al/Ag NPs:a−SiC_0.11_:H/Al foil devices in the flat state and bending state are illustrated in Figure 4b. The I−V characteristics of the device under two kinds of bending radius retain stable flexibility, confirming that the resistive switching behavior is largely unaffected by bending. Figure 4c,e shows the endurance characteristics of the Al/Ag NPs:a−SiC_0.11_:H/Al foil devices after bending with a radius of 10 mm and 6 mm for 1000 cycles of DC sweep, respectively. The stable endurance characteristic of 1000 cycles can be also detected with the on/off ratio consistently remaining around 10^7^. Figure 4d,f displays the retention characteristics the Al/Ag NPs:a−SiC_0.11_:H/Al foil devices after bending with radii of 10 mm and 6 mm, demonstrating stable maintenance of high- and low-resistance states exceeding 10^4^ s. So, the flexible Al/Ag NPs:a−SiC_0.11_:H/Al device is a forming-free resistive memory with an ultra-high on/off ratio, and its resistive switching performance remains unchanged after bending.

The number of bending cycles also serves as a critical parameter for flexible RRAM devices. We investigated the resistive switching behavior of the Al/Ag NPs:a−SiC_0.11_:H/Al foil device after bending for different times with a bending radius of 6 mm. The Al/Ag NPs:a−SiC_0.11_:H/Al foil devices were bent for 100, 500, and 1000 times. Compared with the flat Al/Ag NPs:a−SiC_0.11_:H/Al foil devices, the current values corresponding to the high- and low-resistance states after bending for different times are similar, as shown in Figure 5a. Moreover, variations in forward operating voltage remain within a reasonable fluctuation range. The results demonstrate that the flexible Al/Ag NPs:a−SiC_0.11_:H/Al device can still achieve reversible switching between high- and low-resistance states even after 1000 bending cycles. Figure 5b reveals that the on/off ratio of 10^6^ can be remained from the flexible Al/Ag NPs:a−SiC_0.11_:H/Al foil device after 500 cycles of switching between high- and low-resistance states after a bending cycle of 1000 times, although the high-resistance state exhibits noticeable fluctuations. Endurance characteristic of 10^4^ s substantiate that the high- and low-resistance states stabilize near their original values while retaining a considerably large on/off ratio, as displayed in Figure 5c. To further illustrate the composition of the conductive pathway between the top and bottom electrodes, we have measured the temperature-dependent I-V characteristic of our device, as displayed in Figure 5d. The current intensity of the LRS and HRS shows a slight increase with temperature changes from 240 K to 340 K, revealing the semiconductor conductor characteristics. It is clear that the conductive pathway in our device has no relationship with Ag nanoparticles. It is further illustrated that the formation and rupture of conductive filaments in our device are related to the production of Si dangling bonds and the H^+^ passivation. Therefore, the effect of Joule heating induced by Ag nanoparticles is negligible [39,40,41].

As shown in Figure 6a, multiple resistance states can be observed from the I−V curve of the bent flexible Al/Ag NPs:a−SiC_0.11_:H/Al foil device under various reset voltages. At a reset voltage of −0.5 V, the current decreases by an order of magnitude, achieving the first-level reset state, designated HRS1. As the reset voltage progressively increases to −0.8 V, −1.0 V, and −1.2 V, the output current decreases by an order of magnitude, achieving second−, third−, and fourth−level reset states, respectively, marked as HRS2, HRS3, and HRS4. When the reset voltage reaches −1.2 V, the device undergoes a complete reset, reverting to its HRS state. Through altering the reset voltage, the device transitions sequentially from the LRS state to HRS1, HRS2, HRS3, and ultimately to HRS. Reapplying a set voltage switches the device to the LRS state, allowing for repeated sequential resets. After ten consecutive cycles, the LRS, HRS1, HRS2, HRS3, and HRS states remain distinct, characterized by significant ratios between adjacent resistances with no overlapping, as presented in Figure 6b. It is worth noting that all five resistance states were sustained for over 10^4^ s without degradation, as shown in Figure 6c. The currents corresponding to the high- and low-resistance states of the flexible Al/Ag NPs:a−SiC_0.11_:H/Al device under 50 consecutive positive and negative voltage pulses are displayed in Figure 6d. When the applied pulse amplitude is ±2 V, with a pulse width of 500 ns, both the HRS and LRS states of the device exhibit very stable behavior, maintaining a high switching window ratio of 10^7^. There is no overlap or significant degradation between the high- and low-resistance values. Reducing the pulse width to 100 ns while keeping the amplitude constant results in increased fluctuations in high- and low-resistance values, markedly decreasing the storage window. The switching ratio changes from approximately 10^7^ to 10^4^. Nevertheless, overlap or degradation remains absent, and the device continues to operate normally. Subsequently, the device is first set to the high-resistance state via DC mode. Then, the pulse width is fixed at 200 µs, and the amplitude is increased from 2.0 V to 2.8 V with the step of 0.2 V. Each pulse amplitude is applied nine times continuously, followed by recording the conductance value of the resistance state using a pulse of 0.1 V after each input. It can be observed that the conductance value of the device increases with the increase in the pulse amplitude during the set process. Additionally, at the same amplitude, the greater the pulses’ number is, the higher the conductance value is, as shown in Figure 6e. In contrast, the reset process exhibits an inverse trend, as illustrated in Figure 6f. This reset process also features a fixed pulse width at 200 µs, with amplitude values commencing from −1.4 V to −2.2 V. As the pulse amplitude increases, the conductance value of the device decreases. And when the pulse amplitude remains constant, the greater the pulse number is and the smaller the conductance value is. It is revealed that the flexible device can achieve continuously adjustable conductance under the continuous pulse voltage within a certain amplitude range. Such continuously adjustable conductance characteristics are beneficial for simulating the weight plasticity of biological synapses and have implications in pattern reconstruction and analog signal recording applications [42].

Based on the good consistency and adjustable multilevel resistance states of the flexible Al/Ag NPs:a−SiC_0.11_:H/Al foil resistive switching memory, we explored its neuromorphic characteristics. Twenty-five consecutive alternating positive and negative voltage pulses were applied to the device, repeated for 5 cycles. The pulse width was maintained at 200 µs, with a positive amplitude of 1.5 V and a negative voltage amplitude of −1.5 V. Figure 7a exhibits the schematic illustration of the pre-synapse and the post-synapse based on the Al/Ag NPs:a−SiC_0.11_:H/Al foil device. As shown in Figure 7b, the device’s conductance value gradually increases under continuous positive pulse stimulation, indicating that the positive pulses have an excitatory effect on the synaptic device. Conversely, sustained application of negative pulses leads to a decline in the device’s conductance value, indicating an inhibitory effect of negative pulses on the synaptic device. The results affirm that the flexible Al/Ag NPs:a−SiC_0.11_:H/Al artificial synapse device effectively reproduces the long-term potentiation (LTP) and long-term depression (LTD) functionalities of biological synapses. Figure 7c demonstrates that increasing the electrical pulse voltage from 1.2 V to 1.5 V results in a consistent conductance increase with the number of pulses. The larger the voltage amplitude is, the higher the conductance is. It indicates that the synaptic connection between two neurons is strengthened. In Figure 7d, as the voltage changes from −1.2 V to −1.5 V, the device’s conductance gradually decreases with the number of pulses. The larger the voltage amplitude is, the smaller the conductance is, indicating the process of synaptic inhibition. This process illustrates a weakening of the synaptic connection between two neurons. Spike-timing-dependent plasticity (STDP) is recognized as a sophisticated learning rule within the biological brain and is pivotal for learning and memory [43]. STDP simulation is typically required in evaluating artificial neural networks. Generally, the conductance value of a resistive memory corresponds to the synaptic weight, which can be modified to affect the learning and memory capacities of the synapse. By adjusting the relative timing of pre- and post-synaptic pulses, synaptic weight can be modulated. The change in synaptic weight is defined as:∆W = (I_2_ − I_1_)/I_2_ × 100% (2)
where I_1_ and I_2_ represent the current values before and after the synaptic pulse stimulation, respectively. Figure 7e shows the STDP characteristics of the flexible Al/Ag NPs:a−SiC_0.11_:H/Al foil resistive memory, with the time interval between pre- and post-pulses ranging from 10 µs to 300 µs. It can be observed that the synaptic weight change decreases exponentially with increasing time interval, which is a typical STDP characteristic. The corresponding image memorization simulation of 5 × 5 synaptic arrays based on the flexible Al/Ag NPs:a−SiC_0.11_:H/Al foil device after training of 10, 20, and 30 cycles is displayed in Figure 7f. A good clearness of the synapse arrays to memorize the initial pattern is demonstrated by the simulation maps after 30 iterations. These findings indicate that the flexible artificial synapse device Al/Ag NPs:a−SiC_0.11_:H/Al foil is promising for application in neuromorphic chips.

## 4. Conclusions

In summary, we successfully constructed a flexible artificial synaptic device based on a Ag NPs:a−SiC_0.11_:H memristor by utilizing aluminum foil as the substrate. The device exhibits stable bipolar resistive switching characteristics even after bending for 1000 times, displaying excellent flexibility and uniformity. Furthermore, an on/off ratio of approximately 10^7^ can be obtained. It is found that the incorporation of silver nanoparticles significantly enhances the device’s set and reset voltage uniformity by 76.2% and 69.7%, respectively, which is attributed to the contribution of the Ag nanoparticles. The local electric field of Ag nanoparticles can direct the formation and rupture of conductive filaments. The fitting results of I−V curves show that the carrier transport mechanism agrees with the Poole–Frenkel (P−F) model in the high-resistance state, while the carrier transport follows Ohm’s law in the low-resistance state. Based on the multilevel storage characteristics of the Al/Ag NPs:a−SiC_0.11_:H/Al foil resistive switching device, we successfully observed the biological synaptic characteristics, including long-term potentiation (LTP), long-term depression (LTD), and spike-timing-dependent plasticity (STDP). The flexible artificial Ag NPs:a−SiC_0.11_:H/Al foil synapse possesses excellent conductance modulation capabilities and the visual learning function, demonstrating the promise of application in the flexible electronics technology for high-efficiency neuromorphic computing in the AI period.

## Figures and Tables

**Figure 1 nanomaterials-14-01474-f001:**
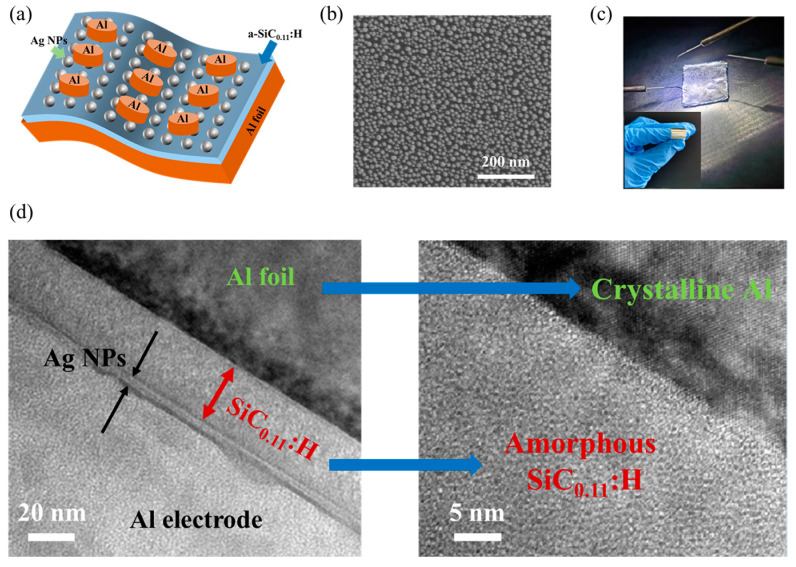
(**a**) Schematic diagram of the Al/Ag NPs:a–SiC_0.11_:H/Al foil device. (**b**) SEM image of Ag NPs:a–SiC_0.11_:H on the surface of Al foil. (**c**) Photograph of the flat Al/Ag NPs:a–SiC_0.11_:H/Al foil device with electrical measurement on the probe station. Inset shows the image of the flexible device. (**d**) X-HRTEM photograph of the Al/Ag NPs:a–SiC_0.11_:H/Al foil device after set process.

**Figure 2 nanomaterials-14-01474-f002:**
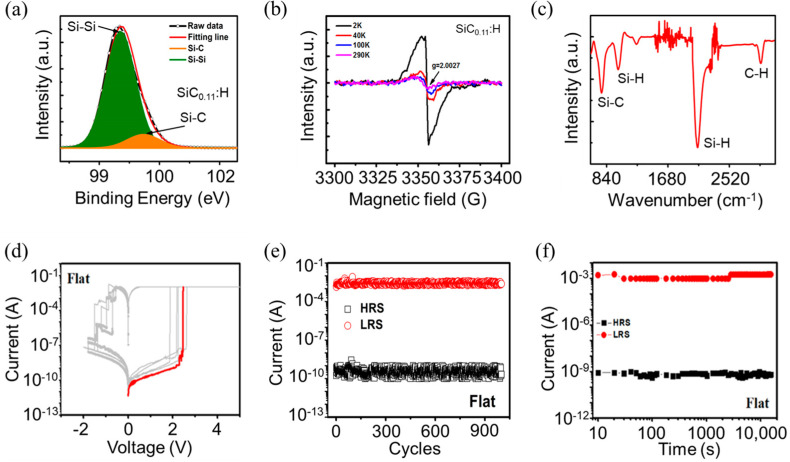
(**a**) XPS spectra of the SiC_0.11_:H films. (**b**) ESR spectra of the SiC_0.11_:H films. (**c**) FTIR spectra of the SiC_0.11_:H films. (**d**) I–V curves of the flat Al/Ag NPs:a–SiC_0.11_:H/Al foil device. (**e**) Endurance characteristics of the flat Al/Ag NPs:a–SiC_0.11_:H/Al foil device. (**f**) Retention characteristics of the flat Al/Ag NPs:a–SiC_0.11_:H/Al foil device at room temperature.

**Figure 3 nanomaterials-14-01474-f003:**
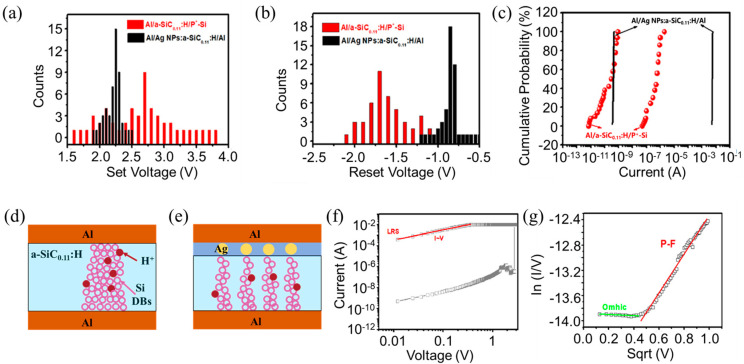
(**a**,**b**) The switching voltage distribution of the Al/a−SiC_0.11_:H/P^+^−Si and the flat Al/Ag NPs:a−SiC_0.11_:H/Al device after 50 cycles, respectively. (**c**) Accumulative probability of LRS/HRS state of the Al/a−SiC_0.11_:H/P^+^−Si and the flat Al/Ag NPs:a−SiC_0.11_:H/Al device after 50 cycles, respectively. (**d**,**e**) Diagram of the resistive switching mechanism for the Al/a−SiC_0.11_:H/Al and Al/Ag NPs:a−SiC_0.11_:H/Al devices. (**f**) The I−V curves of the set process of the flat Al/Ag NPs:a−SiC_0.11_:H/Al device. (**g**) In the HRS state, the P−F fitting of the I−V curve of the flat Al/Ag NPs:a−SiC_0.11_:H/Al device.

**Figure 4 nanomaterials-14-01474-f004:**
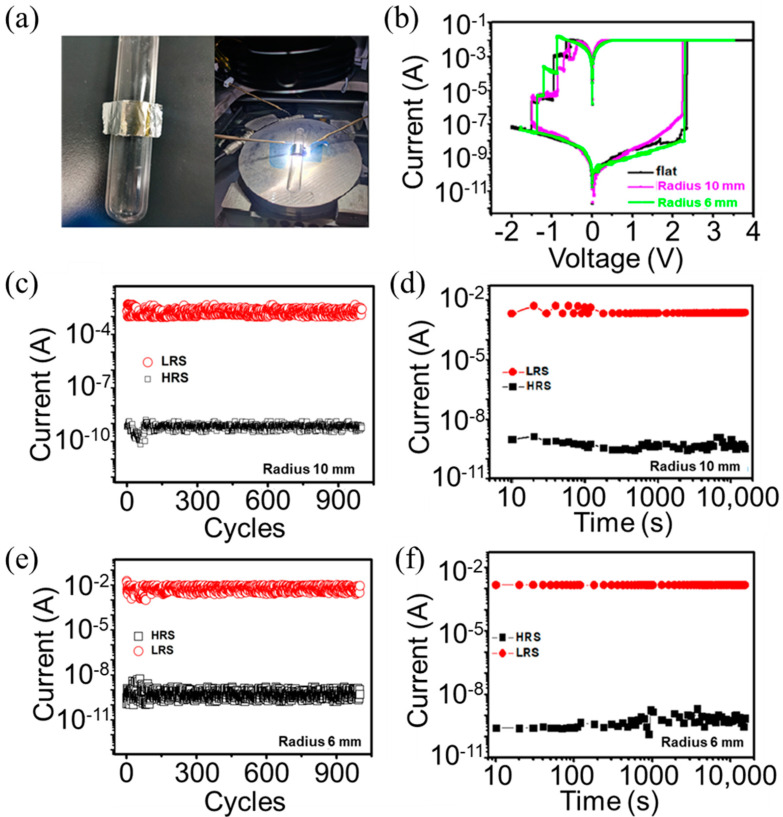
(**a**) The image and the electrical measurement of the flexible Al/Ag NPs:a−SiC_0.11_:H/Al foil device around a glass cylinder. (**b**) The I−V curves of the flat Al/Ag NPs:a−SiC_0.11_:H/Al foil device and the flexible Al/Ag NPs:a−SiC_0.11_:H/Al device around a glass cylinder with different bending radii. (**c**,**d**) The endurance and retention characteristics of the flexible Al/Ag NPs:a-SiC_0.11_:H/Al foil device around a glass cylinder with a bending radius of 10 mm. (**e**,**f**) The endurance and retention characteristics of the flexible Al/Ag NPs:a−SiC_0.11_:H/Al device around a glass cylinder with a bending radius of 6 mm.

**Figure 5 nanomaterials-14-01474-f005:**
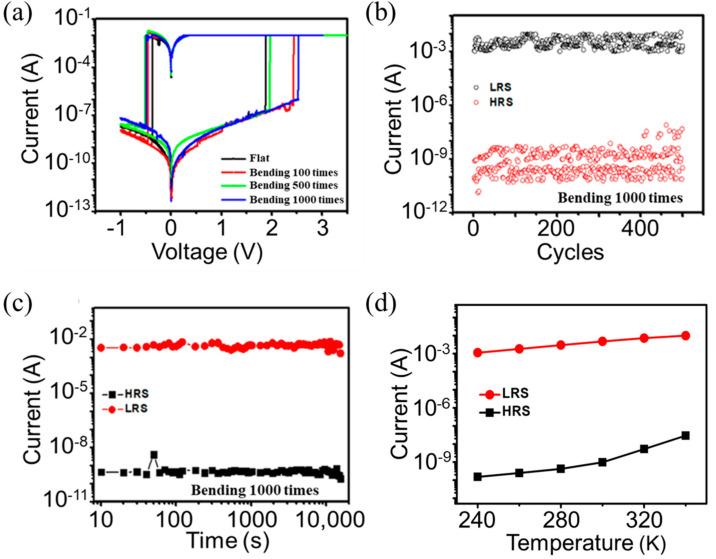
The flexible Al/Ag NPs:a−SiC_0.11_:H/Al foil device is wrapped around a glass cylinder with a bending radius of 6 mm. (**a**) The I−V curves of the flexible device after different repetitions of bending. (**b**) The endurance characteristics of the flexible device after 1000 repetitions of bending. (**c**) The retention characteristics of the flexible device after 1000 repetitions of bending. (**d**) The temperature-dependent I−V curves of the flexible device in HRS and LRS.

**Figure 6 nanomaterials-14-01474-f006:**
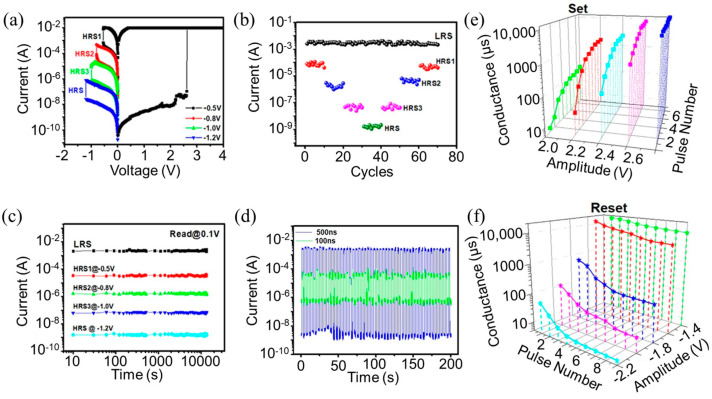
The flexible Al/Ag NPs:a−SiC_0.11_:H/Al foil device is wrapped around a glass cylinder with a bending radius of 6 mm. (**a**) The I−V curves of the flexible device under different reset voltages. (**b**) The endurance characteristics of the flexible device corresponding to five resistive switching states. (**c**) The retention characteristics of the flexible device corresponding to five resistive switching states. (**d**) The output current of the flexible device when applying 50 continuous positive and negative electrical pulses. (**e**,**f**) The conductivity characteristics of the flexible device in set and reset processes when applying electrical pulses of different amplitude (different color line).

**Figure 7 nanomaterials-14-01474-f007:**
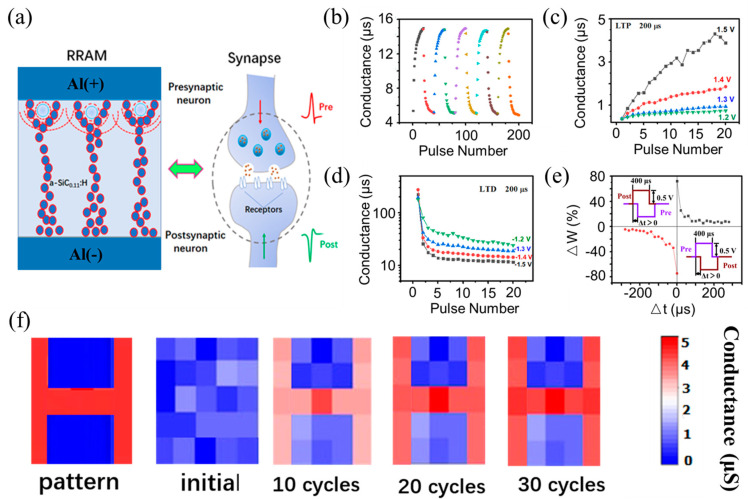
The flexible Al/Ag NPs:a−SiC_0.11_:H/Al foil device is wrapped around a glass cylinder with a radius of 6 mm. (**a**) A diagrammatic of Al/Ag NPs:a−SiC_0.11_:H/Al synaptic plasticity between the presynaptic neuron and the postsynaptic neuron. (**b**) The conductance changes of the flexible device when 25 positive electrical pulses were applied, followed by 25 negative electrical pulses. (**c**) The LTP characteristics of the flexible device when applying positive pulses with different amplitudes. (**d**) The LTD characteristics of the flexible device when applying negative pulses with different amplitudes. (**e**) The STDP characteristics of the flexible synapse device. (**f**) The simulation of image memorization corresponding to the 5 × 5 synaptic arrays after training for 10, 20, and 30 iterations.

## Data Availability

The original contributions presented in the study are included in the article, further inquiries can be directed to the corresponding author.

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
