# Peer review of "Flexible Artificial Ag NPs:a–SiC0.11:H Synapse on Al Foil with High Uniformity and On/Off Ratio for Neuromorphic Computing"

_nanomaterials, 2024, doi:10.3390/nano14181474_

Round 1

Reviewer 1 Report

Comments and Suggestions for Authors

Flexible Artificial Ag NPs:a-SiC0.11:H Synapse on Al Foil with High Uniformity and On/Off Ratio for Neuromorphic Computing

Judgement: Minor Revision

Summary: In this study, the authors report a flexible artificial synaptic device based on Ag NPs:a-SiC0.11:H memristor constructed on aluminium foil substrate. The device exhibits stable bipolar resistive switching characteristics with excellent on-off ratio, retention and good mechanical flexibility. But the endurance performance is very poor compared to literature and needs improvement to warrant publication. The stability of the RRAM performance is attributed to the presence of Ag nanoparticles that aids the local electric field in the device. Based on the multilevel storage characteristics of the Al/Ag NPs:a-SiC0.11:H/Al RRAM, the authors successfully demonstrate spike-timing-dependent plasticity (STDP) and other neuromorphic functionalities, useful for computing..

The authors should also address the following comments before publication.

Comments:

1.    The endurance performance of 100 cycles is very poor compared to literature. The authors are recommended to improve this metric and submit a revised manuscript version for review.

2.    The authors show that Ag nanoparticles are essential to achieving the excellent performance. Would it be possible to image the filament formation and rupture through electron microscopy for samples with and without Ag nanoparticles to prove this point explicitly?

3.    Have the authors considered the effect of Joule heating that maybe modified with nanoparticles as supposed to pure thin films? Eg in literature: Nat Commun 13, 2074 (2022). https://doi.org/10.1038/s41467-022-29727-1

Reviewer 2 Report

Comments and Suggestions for Authors

The authors have presented the paper entitled "Flexible Artificial Ag NPs:a-SiC0.11:H Synapse on Al Foil with High Uniformity and On/Off Ratio for Neuromorphic Computing" 

In general the paper is very interesting and it has enormous applications. However I have several questions.

1.- The authors claim that they have got an a-SiC on topo of an aluminum foil. However, there are not XRD to probe that the SIC that they have got is and a-SiC and also if so with phase or if is amorphous or 3C, 4H, 6H kind of crystalline structure. Can you comment on that , please?

2.- The SiC signal on the XPS is very small, even I m not sure is there. Why is that the authors got such an small vs high intensity Si signal? Why is the Si 2p from Silicon even there?

3.- It is difficult to believe that Infrared spectrum got a SiC signal (due to the small amount, from XPS) , it seems to be more of a Si-H signal, and also some other signals look a bit our of place, specially since the spectra x axis values are so difficult to get accurately.

4.- From the NPs , no spectra or images are shown, please provide.

5.- Do you have any image of the NPs arranged after deposit?

6.- How can you correlate all of these with your results?

Thanks  for the time.

Comments on the Quality of English Language

English can be improve, it seems there are some general mistakes in the way of writing. As an example the beginning of the Materials and Methods.

Round 2

Reviewer 2 Report

Comments and Suggestions for Authors

The authors have reviewed and changed the paper and I consider that it has been improved greatly. They have answer all the questions asked and the paper I consider now is ready to be published.